# Fairness Transferability
# Subject to Bounded Distribution Shift

**Yatong Chen**[†]**, Reilly Raab**[†]**, Jialu Wang, Yang Liu**[∗]

University of California, Santa Cruz
{ychen592, reilly, faldict, yangliu}@ucsc.edu

## Abstract

Given an algorithmic predictor that is "fair" on some *source* distribution, will it still be fair on an unknown *target* distribution that differs from the source within some bound? In this paper, we study the *transferability of statistical group fairness* for machine learning predictors (*i.e.*, classifiers or regressors) subject to bounded distribution shift. Such shifts may be introduced by initial training data uncertainties, user adaptation to a deployed predictor, dynamic environments, or the use of pre-trained models in new settings. Herein, we develop a bound that characterizes such transferability, flagging potentially inappropriate deployments of machine learning for socially consequential tasks. We first develop a framework for bounding violations of statistical fairness subject to distribution shift, formulating a generic upper bound for transferred fairness violations as our primary result. We then develop bounds for specific worked examples, focusing on two commonly used fairness definitions (*i.e.*, demographic parity and equalized odds) and two classes of distribution shift (*i.e.*, covariate shift and label shift). Finally, we compare our theoretical bounds to deterministic models of distribution shift and against real-world data, finding that we are able to estimate fairness violation bounds in practice, even when simplifying assumptions are only approximately satisfied.

## 1 Introduction

*Distribution shift* is a common, real-world phenomenon that affects machine learning deployments when the *target* distribution of examples (features and labels) ultimately encountered by a data-driven policy diverges from the *source* distribution it was trained for. For socially consequential decisions guided by machine learning, such shifts in the underlying distribution can invalidate fairness guarantees and cause harm by exacerbating social disparities. Unfortunately, distribution shift can be technically difficult or impossible to model at training time (*e.g.*, when depending on complex social dynamics or unrealized world events). Nonetheless, we still wish to certify the robustness of fairness metrics for a policy on possible target distributions.

In this paper, we provide a general framework for quantifying the robustness of statistical group fairness guarantees. We assume that the target distribution is adversarially drawn from a bounded domain, thus reducing the hard problem of modelling distribution shift dynamics to a more tractable, static problem. With this framework, we can detect potentially inappropriate policy applications, prior to deployment, when fairness violation bounds are not sufficiently small.

This work bridges a gap between recent literature on *domain adaptation*, which has largely focused on the transferability of prediction accuracy (rather than fairness), and *algorithmic fairness*, which

---

[†]These authors contributed equally to this work.
[∗]Corresponding author: yangliu@ucsc.edu

has typically considered static distributions or prescribed models of distribution shift. Our work is the first to systematically bound quantifiable violations of statistical group fairness while remaining agnostic to (1) the mechanisms responsible for distribution shift, (2) how group-specific distribution shifts are quantified, and (3) the specific statistical definition of group fairness applied.

Our primary result is a bound on a policy's potential "violation of statistical group fairness"—defined in terms of the differences in policy outcomes between groups—when applied to a target distribution shifted relative to the source distribution within known constraints. Such settings naturally arise whenever training data represents a random sample of a target population with different statistics or a sample from dynamic environments, when a policy is reused on a new distribution without retraining, or whenever policy deployment itself induces a distribution shift. As an example of this last case, strategic individuals seeking loans might change their features or abstain from future application (thus shifting the distribution of examples) in response to policies trained on historical data [18, 38, 43]. Beyond policy selection, exogenous pressure such as economic trends and noise may also drive distribution shift in this example.

In Figure 1, we show how a real-world distribution shift in demographic and income data for US states between 2014 and 2018 may increase fairness violations while decreasing accuracy for a hypothetical classifier

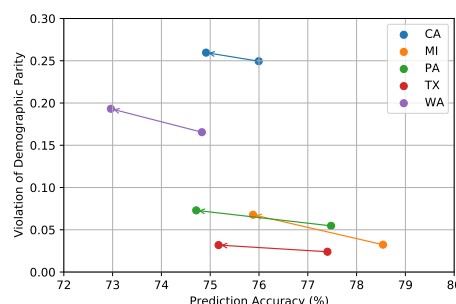

**Figure 1:** In Section 7, we evaluate our bounds against historical, temporal distribution shifts in demographics and income recorded by the US Census Bureau [13]. The above figure depicts changes to income-prediction accuracy and demographic parity violation when a classifier initially trained on US state-specific demographic data for 2014 is reused on 2018 data, thus exemplifying the negative potential effects of distribution shift.

trained on the 2014 distribution. In such settings, it is useful to quantify how fairness guarantees transfer across distributions shifted within some bound, thus allowing the deployment of unfair machine learning policies to be avoided.

## 1.1 Related Work

Our work considers a setting similar to recent studies of *domain adaptation*, which have largely focused on characterizing the effects of distribution shift on prediction performance rather than fairness. Our work also builds on efforts in *algorithmic fairness*, especially *dynamical* treatments of distribution shift in response to deployed machine learning policies [23, 11, 30]. We reference specific prior work in these domains in Appendix B, and here discuss existing work that focuses on how certain measures of fairness are affected when policies are subject to specific distribution shift.

**Fairness subject to Distribution Shift:** A number of recent studies have considered specific examples of fairness transferability subject to distribution shift [34, 9, 36, 31, 21]. In particular, Schumann et al. [34] examine *equality of opportunity* and *equalized odds* as definitions of group fairness subject to distribution shifts quantified by an $\mathcal{H}$-divergence function; Coston et al. [9] consider *demographic parity* subject to a *covariate shift* assumption while group identification remains unavailable to the classifier; Singh et al. [36] focus on common group fairness definitions for binary classifiers subject to a class of distribution shift that generalizes covariate shift and label shift by preserving some conditional probability between variables; and Rezaei et al. [31] similarly consider common binary classification fairness definitions such as equalized odds subject to covariate shift. While we address similar settings to these works as special cases of our bound, we propose a unifying formulation for a broader class of statistical group fairness definitions and distribution shifts. In doing so, we recognize that particular settings recommend themselves to more natural measures of distribution shift, providing examples in Section 4.1, Section 4.2, and Section 5).

Another thread in existing literature is the development of robust models with the goal of guaranteeing fairness on a modelled target distribution (e.g., [1, 32, 26, 3, 21]) , for example, by assuming covariate shift and the availability of some unlabelled target data [9, 36, 31]. In particular, Singh et al. [36] focus on learning stable models that will preserve prediction accuracy and fairness, utilizing a causal graph to describe anticipated distribution shifts. Rezaei et al. [31] takes a robust optimization approach, and Coston et al. [9] develops prevalence-constrained and target-fair covariate shift method for getting the robust model. In contrast, our goal is to quantify fairness violations after an *adversarial* distribution shift for *any* given policy, including those not trained with robustness in mind.

## 1.2 Our Contributions

Our primary contribution is formulating a general, worst-case upper bound for a given policy's violation of statistical group fairness subject to group-dependent distribution shifts within presupposed bounds (i.e., Equation (9)). Bounding violations of fairness subject to distribution shift allows us to recognize and avoid potentially inappropriate deployments of machine learning when the potential disparities of a prospective policy eclipse a given threshold within bounded distribution shifts of the training distribution.

We first characterize the space of statistical group fairness definitions and possible distribution shifts by appeal to premetric functions (Definition 2.1). After formulating the worst-case upper bound, we explore common sets of simplifying assumptions for this bound as special cases, yielding tractable calculations for several familiar combinations of fairness definitions and subcases of distribution shift (Theorem 4.1, Theorem 5.2) with readily interpretable results. Finally, we compare our theoretical bounds to prescribed models of distribution shift in Section 6 and to real-world data in Section 7. The details for reproducing our experimental results can be found at
`https://github.com/UCSC-REAL/Fairness_Transferability`.

## 2 Formulation

*The appendices include a table of notation (Appendix A) and all proofs (Appendix F).*

### 2.1 Algorithmic Prediction

We consider two distributions, $\mathcal{S}$ (*source*) and $\mathcal{T}$ (*target*), each defined as a probability distribution for *examples*, where each example defines values for three random variables: $X$, a **feature** (*e.g.*, $x$) with arbitrary domain $\mathcal{X}$; $Y$, a **label** (*e.g.*, $y$) with arbitrary domain $\mathcal{Y}$; and $G$, a **group** (*e.g.*, $g$ or $h$) with finite, countable domain $\mathcal{G}$. The predictor's policy $\pi$, intended for $\mathcal{S}$ but used on $\mathcal{T}$, defines a fourth variable for each example: *viz.*, $\hat{Y}$, a **predicted label** (*e.g.*, $\hat{y}$) with domain $\hat{\mathcal{Y}} = \mathcal{Y}$.

Using $\mathcal{P}(\cdot)$ to denote the space of probability distributions over some domain, we denote the space of distributions over examples as $\mathbb{D} := \mathcal{P}(\mathcal{X} \times \mathcal{Y} \times \mathcal{G})$, such that $\mathcal{S}, \mathcal{T} \in \mathbb{D}$. It will also be useful for us to notate the space of distributions over example *outcomes* associated with a given policy as $\mathbb{O} := \mathcal{P}(\mathcal{X} \times \mathcal{Y} \times \hat{\mathcal{Y}})$ and the space of distributions over of *group-specific examples* as $\mathbb{G} := \mathcal{P}(\mathcal{X} \times \mathcal{Y})$.

Without loss of generality, we allow the prediction policy $\pi$ to be stochastic, such that, for any combination $(x, g)$, the predictor effectively samples $\hat{Y}$ from a corresponding probability distribution $\pi(x, g)$. Stochastic classifiers arise in various constrained optimization problems and proven useful for making problems with custom losses or fairness constraints tractable [10, 17, 29, 40].

We denote the space of nondeterministic policies as $\Pi := (\mathcal{X} \times \mathcal{G} \to \mathcal{P}(\hat{\mathcal{Y}}))$ (*e.g.*, $\pi \in \Pi$) and utilize the natural transformations that relate the spaces of distributions $\mathbb{D}$, policies $\Pi$, and outcomes $\mathbb{O}$:

$$\Pr_{\pi,\mathcal{T}}(\hat{Y}=\hat{y}, X=x, G=g) = \Pr_{\hat{Y} \sim \pi(x,g)}(\hat{Y}=\hat{y}) \cdot \Pr_{X,G \sim \mathcal{T}}(X=x, G=g) \tag{1}$$

We abuse the $\Pr$ notation for both probability *density* and probability *mass* functions as appropriate.

### 2.2 Statistical Group-Fairness

We next define a broad class of *disparity* functions $\Delta^{\star} \colon \Pi \times \mathbb{D} \to \mathbf{R}$ representing how "unfair" a given policy is for a given distribution (*e.g.*, writing $\Delta^{\star}(\pi, \mathcal{T})$), noting that this notion of fairness is limited to capturing statistical discrepancies of outcomes between groups.

**Definition 2.1.** We define a *premetric*[3] $\Psi$ on the space of distributions $p$ with respect to $q$ by the properties $\Psi(p \parallel q) \geq 0$ and $\Psi(p \parallel p) = 0$ for all $p, q$, and refer to the value of $\Psi$ as a "shift".

**Definition 2.2.** We define a *statistical group disparity* $\Delta^{\star}$ for policy $\pi$ and distribution $\mathcal{T}$ in terms of the symmetrized shifts between group-specific outcome distributions. We measure shifts between outcome distributions with a given premetric $\Psi \colon \mathbb{O}^2 \to \mathbf{R}$.

$$\Delta^{\star}(\pi, \mathcal{T}) := \sum_{g,h \in \mathcal{G}} \Psi\Big( \Pr_{\pi,\mathcal{T}}(X, Y, \hat{Y} \mid G=g) \parallel \Pr_{\pi,\mathcal{T}}(X, Y, \hat{Y} \mid G=h) \Big) \tag{2}$$

---

[3]Despite use on Wikipedia, this is not a standard term in the literature. In general, the axioms of a premetric as defined in Definition 2.1 are a subset (thus "pre") of those that define a metric.

In Definition 2.2, $\Psi$ quantifies the specific statistical differences in outcomes between groups that are "unfair", where a value of 0 implies perfect fairness. In this work, we assume that $\Psi$ is the same for all $g, h$ and that $\Delta^\star$ is insensitive to relative group size $\Pr(G)$.

**Examples** Familiar applications of Definition 2.2 include *demographic parity* (DP) and *equalized odds* (EO). A policy satisfying DP, in expectation, assigns a given binary classification $y \in \{0, 1\}$ to the same fraction of examples in each group. We may measure the violation of DP as

$$\Delta^\star_{\mathsf{DP}}(\pi, \mathcal{T}) \coloneqq \sum_{g,h \in \mathcal{G}} \left| \Pr_{\pi,\mathcal{T}}(\hat{Y}{=}1 \mid G{=}g) - \Pr_{\pi,\mathcal{T}}(\hat{Y}{=}1 \mid G{=}h) \right| \tag{3}$$

The associated *premetric* $\Psi_{\mathsf{DP}}$ for $p, q \in \mathbb{O}$ is $\Psi_{\mathsf{DP}}(p \parallel q) = \left| \Pr_p(\hat{Y}{=}1) - \Pr_q(\hat{Y}{=}1) \right|$.

To satisfy EO, for binary $\mathcal{Y} = \{0, 1\}$, $\pi$ must maintain group-invariant true positive and false positive classification rates. We may measure the violation of EO as

$$\Delta^\star_{\mathsf{EO}}(\pi, \mathcal{T}) \coloneqq \sum_{g,h \in \mathcal{G}} \sum_{y \in \mathcal{Y}} \left| \Pr_{\pi,\mathcal{T}}(\hat{Y}{=}1 \mid G{=}g, Y{=}y) - \Pr_{\pi,\mathcal{T}}(\hat{Y}{=}1 \mid G{=}h, Y{=}y) \right| \tag{4}$$

The associated premetric is $\Psi_{\mathsf{EO}}(p \parallel q) = \sum_{y \in \mathcal{Y}} \left| \Pr_p(\hat{Y}{=}1 \mid Y{=}y) - \Pr_q(\hat{Y}{=}1 \mid Y{=}y) \right|$. Note that the restriction of EO to the $(Y = 1)$ case is known as *Equal Opportunity* (EOp).

We remark that Definition 2.2 provides a unifying representation for a wide array of statistical group "unfairness" definitions and may be used with inequality constraints. That is, we may recover many working definitions of fairness that effectively specify a maximum value of disparity:

**Definition 2.3.** A policy $\pi$ is $\epsilon$-**fair** with respect to $\Delta^\star$ on distribution $\mathcal{T}$ iff $\Delta^\star(\pi, \mathcal{T}) \leq \epsilon$.

### 2.3 Vector-Bounded Distribution Shift

Suppose, after developing policy $\pi$ for distribution $\mathcal{S}$, we realize some new distribution $\mathcal{T}$ on which the policy is actually operating. This realization may be the consequence of sampling errors during the learning process, strategic feedback to our policy, random processes, or the reuse of our policy on a new distribution for which retraining is impractical. Our goal is to bound $\Delta^\star(\pi, \mathcal{T})$ given knowledge of $\Delta^\star(\pi, \mathcal{S})$ and some notion of how much $\mathcal{T}$ possibly differs from $S$.

**Definition 2.4.** $K(p \parallel q)$ is a *divergence* if and only if for all $p$ and $q$, $K(p \parallel q) \geq 0$ and $K(p \parallel q) = 0 \iff q = p$.

**Definition 2.5.** Define the group-vectorized shift $\mathbf{D}$, as $\mathcal{S}$ mutates into $\mathcal{T}$, as

$$\mathbf{D}(\mathcal{T} \parallel \mathcal{S}) \coloneqq \sum_g \mathbf{e}_g D_g \left( \Pr_{\mathcal{T}}(X, Y \mid G{=}g) \parallel \Pr_{\mathcal{S}}(X, Y \mid G{=}g) \right) \tag{5}$$

where $\mathbf{e}_g$ represents a unit vector indexed by $g$, and each $D_g \colon \mathbb{G}^2 \to \mathbf{R}$ is a divergence (Definition 2.4). Note that each $D_g$ also defines a premetric (but not necessarily a divergence) on $\mathbb{D}$.

**Assumption 2.6.** Let there exist some vector $\mathbf{B} \succeq 0$ bounding $\mathbf{D}(\mathcal{T} \parallel \mathcal{S}) \preceq \mathbf{B}$, where $\preceq$ and $\succeq$ denote element-wise inequalities.

In Assumption 2.6, $\mathbf{B}$ limits the possible distribution shift as $\mathcal{S}$ mutates into $\mathcal{T}$, without requiring us to specify a model for how distributions evolve. When modelling distribution shift requires complex dynamics (*e.g.*, when agents learn and respond to classifier policy), we reduce a potentially difficult dynamical problem to a more tractable, adversarial problem to achieve a bound.

**Lemma 2.7.** *For all $\pi$, $\Delta^\star$, and $\mathbf{D}$, when $\mathbf{B} = 0$, $\Delta^\star(\pi, \mathcal{S}) = \Delta^\star(\pi, \mathcal{T})$.*

Lemma 2.7 indicates that, for a fixed policy $\pi$, a change in disparity requires a measurable shift in distributions from $\mathcal{S}$ to $\mathcal{T}$, confirming intuition.

**Restricted Distribution Shift** Common assumptions that restrict the set of distribution shifts include *covariate shift* and *label shift*. For covariate shift, the distribution of *labels* conditioned on *features* is preserved across distributions for all groups, while for label shift, the distributions of *features* conditioned on *labels* is preserved across distributions for all groups.

$$\textbf{Covariate shift} \text{ implies } \Pr_{\mathcal{T}}(Y \mid X, G) = \Pr_{\mathcal{S}}(Y \mid X, G) \tag{6}$$

$$\textbf{Label shift} \text{ implies } \Pr_{\mathcal{T}}(X \mid Y, G) = \Pr_{\mathcal{S}}(X \mid Y, G) \tag{7}$$

In Section 4, we explore a deterministic model of a population's response to classification as an example of covariate shift. We do the same in Section 5 for label shift.

## 3 General Bounds

We first define a primary bound in Definition 3.1 before considering simplifying special cases.

Given an element-wise bound $\mathbf{B}$ on the vector-valued shift $\mathbf{D}(\mathcal{T} \parallel \mathcal{S})$ (Assumption 2.6) we may bound the disparity $\Delta^\star$ of policy $\pi$ on any realizable target distribution $\mathcal{T}$ by its supremum value.

**Definition 3.1.** Define the supremum value $v$ for $\Delta^\star$ subject to $\mathbf{D}(\mathcal{T} \parallel \mathcal{S}) \preceq \mathbf{B}$ as

$$v(\Delta^\star, \mathbf{D}, \pi, \mathcal{S}, \mathbf{B}) \coloneqq \sup_{\mathbf{D}(\mathcal{T}\parallel\mathcal{S})\preceq\mathbf{B}} \Delta^\star(\pi, \mathcal{T}) \tag{8}$$

$$\mathbf{D}(\mathcal{T} \parallel \mathcal{S}) \preceq \mathbf{B} \implies \Delta^\star(\pi, \mathcal{T}) \leq v(\Delta^\star, \mathbf{D}, \pi, \mathcal{S}, \mathbf{B}) \tag{9}$$

In general, our strategy is to exploit the mathematical structure of the setting encoded by $\Delta^\star$ (*i.e.*, $\Psi$) and $\mathbf{D}$ to obtain an upper bound for $v$ defined in Equation (8). We first explore general cases of simplifying assumptions before presenting worked special examples for frequently encountered settings. Finally, we compare the resulting theoretical bounds to numerical results and simulations.

### 3.1 Lipshitz Conditions

The value of $v$ defines a scalar field in $\mathbf{B}$ and therefore a conservative vector field $\mathbf{F} = \nabla_{\mathbf{B}} v$.

For any curve in $\mathbb{D}$ from $\mathcal{S}$ to $\mathcal{T}$, bounds of the form $\mathbf{F} \preceq \mathbf{L}$ for some constant $\mathbf{L}$ along the curve imply a Lipshitz bound on $\Delta^\star$. We visualize a bound in Figure 2 for all possible curves in the region $\mathbf{D}(\mathcal{T} \parallel \mathcal{S}) \preceq \mathbf{B}$.

**Theorem 3.2** (Lipshitz Upper Bound). *If there exists an* $\mathbf{L}$ *such that* $\nabla_{\mathbf{b}} v(\Delta^\star, \mathbf{D}, \pi, \mathcal{S}, \mathbf{b}) \preceq \mathbf{L}$, *everywhere along some curve as* $\mathbf{b}$ *varies from* $0$ *to* $\mathbf{B}$, *then*

$$\Delta^\star(\pi, \mathcal{T}) \leq \Delta^\star(\pi, \mathcal{S}) + \mathbf{L} \cdot \mathbf{B} \tag{10}$$

Succinctly, if we are guaranteed that disparity can never increase faster than a certain rate in some measure of distribution shift, then, given a maximum distribution shift, this rate bounds the maximum possible disparity. The utility of Theorem 3.2 arises when a Lipshitz condition $\mathbf{L}$ is known, but direct computation of $v$ is difficult. We provide an example of a Lipshitz bound in Section 5.

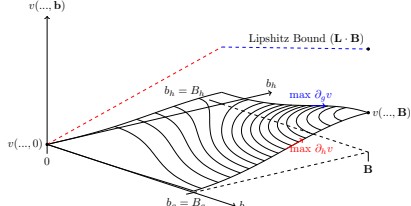

**Figure 2:** A Lipshitz bound for all curves parameterized by distribution shift bound $\mathbf{b}$ in the $(0, \mathbf{B})$ $\mathbb{D}$-hyperrectangle on the surface $v$. In the figure, for groups $i \in \{g, h\}$, $\max \partial_i v = L_i$, and the colored dotted lines corresponds to $L_i b_i$, which, when summed, equal $\mathbf{L} \cdot \mathbf{B}$.

### 3.2 Subadditivity Conditions

**Definition 3.3.** Define $w$ as the maximum *increase* in disparity subject to $\mathbf{D}(\mathcal{T} \parallel \mathcal{S}) \preceq \mathbf{B}$, *i.e.*, $w(\Delta^\star, \mathbf{D}, \pi, \mathcal{S}, \mathbf{B}) \coloneqq v(\Delta^\star, \mathbf{D}, \pi, \mathcal{S}, \mathbf{B}) - \Delta^\star(\pi, \mathcal{S})$.

**Theorem 3.4.** *Suppose, in the region* $\mathbf{D}(\mathcal{T} \parallel \mathcal{S}) \preceq \mathbf{B}$, *that $w$ is subadditive in its last argument. That is,* $w(..., \mathbf{a}) + w(..., \mathbf{c}) \geq w(..., \mathbf{a} + \mathbf{c})$ *for* $\mathbf{a}, \mathbf{c} \succeq 0$ *and* $\mathbf{a} + \mathbf{c} \preceq \mathbf{B}$. *If $w$ is also locally differentiable, then a first-order approximation of $w(..., \mathbf{b})$ evaluated at $0$, i.e.,*

$$\mathbf{L} = \nabla_{\mathbf{b}} w(..., \mathbf{b})\big|_{\mathbf{b}=0} = \nabla_{\mathbf{b}} v(..., \mathbf{b})\big|_{\mathbf{b}=0} \tag{11}$$

*provides an upper bound for* $v(..., \mathbf{B})$, *i.e.,*

$$v(\Delta^\star, \mathbf{D}, \pi, \mathcal{S}, \mathbf{B}) \leq \Delta^\star(\pi, \mathcal{S}) + \mathbf{L} \cdot \mathbf{B} \tag{12}$$

Theorem 3.4 notes that "diminishing returns" in the change of $\Delta^\star$ as the difference of $\mathcal{T}$ with respect to $\mathcal{S}$ is increased implies a bound on $\Delta^\star$ in terms of its local sensitivity to $\mathbf{D}$ at $\mathcal{S}$ (*i.e.*, using a first-order Taylor approximation). Note that, if $w$ is concave in the bounded region, it is also subadditive in the bounded region, but the converse is not true, nor does the converse imply Lipshitzness.

## 3.3 Geometric Structure

It may happen that $\Psi \colon \mathbb{O}^2 \to \mathbf{R}$ and each $D_g \colon \mathbb{G}^2 \to \mathbf{R}$ share structure that permits a geometric interpretation of distribution shift. While the utility of this observation depends on the specific properties of $\Psi$ and $\mathbf{D}$, we demonstrate a worked example building on Section 4.2 in Appendix D, in which we allow ourselves to select a suitable $\mathbf{D}$ for ease of interpretation. We proceed to consider worked examples that adopt common assumptions limiting the form of distribution shift and apply common definitions of statistical group fairness.

## 4 Covariate Shift

We now present our fairness transferability results subject to covariate shift for both demongraphic parity (Section 4.1) and equalized opportunity (Section 4.2) as fairness criteria.

### 4.1 Demographic Parity

The simplest way to work with Equation (8) is to bound the supremum $v$. We first consider demographic parity (Equation (3)) for $\mathcal{Y} = \{0, 1\}$ and $\mathcal{G} = \{g, h\}$, subject to covariate shift (Equation (6)). We find that the form of $\Delta^\star_{\mathsf{DP}}$ subject to covariate shift recommends itself to a natural choice of vector divergence, $\mathbf{D}$. First, define a *re-weighting coefficient* $\omega_g(\mathcal{T}, \mathcal{S}, x) \coloneqq \frac{\Pr_{\mathcal{T}}(X=x \mid G=g)}{\Pr_{\mathcal{S}}(X=x \mid G=g)}$.

**Theorem 4.1.** *For demographic parity between two groups under covariate shift (denoting, for each $g$, $\beta_g \coloneqq \Pr_{\pi, \mathcal{S}}(\hat{Y}=1 \mid G=g)$),*

$$\Delta^\star_{\mathsf{DP}}(\pi, \mathcal{T}) \leq \Delta^\star_{\mathsf{DP}}(\pi, \mathcal{S}) + \sum_g \left( \beta_g(1 - \beta_g) \cdot \mathrm{Var}_{\mathcal{S}}[\omega_g(\mathcal{T}, \mathcal{S}, x)] \right)^{1/2} \tag{13}$$

We notice that $\mathrm{Var}_{\mathcal{S}}[\omega_g(\mathcal{T}, \mathcal{S}, x)]$ recommends itself as a suitable divergence $D_g$ from $\mathcal{S}$ to $\mathcal{T}$. Using basis vectors $\mathbf{e}_g$, for this example, we could define $\mathbf{D}(\mathcal{T} \parallel \mathcal{S}) = \sum_g \mathbf{e}_g \mathrm{Var}_{\mathcal{S}}[\omega_g(\mathcal{T}, \mathcal{S}, x)]$. When $\mathrm{Var}_{\mathcal{S}}[\omega_g(\mathcal{T}, \mathcal{S}, x)] \leq B_g$, it follows $\Delta^\star_{\mathsf{DP}}(\pi, \mathcal{T}) \leq \Delta^\star_{\mathsf{DP}}(\pi, \mathcal{S}) + \sum_g \left( \beta_g(1 - \beta_g) \cdot B_g \right)^{1/2}$. Comparing the inequality in Theorem 4.1 and the consequent of Equation (10), we can interpret $\Pr_{\pi, \mathcal{S}}(\hat{Y}=1)$ in Theorem 4.1 as an upper bound for the *average* value of $\nabla_{\mathbf{b}} v(\Delta^\star_{\mathsf{DP}}, \mathbf{D}, \pi, \mathcal{T}, \mathbf{b})$ along any curve from $\mathcal{S}$ to $\mathcal{T}$. Interpreting this result, the closer $\Pr(\hat{Y}=1)$ is to 0.5 for any group, the more potentially sensitive the fairness of the policy is to distribution shifts for that group. We can further generalize the results to multi-class and multi-group setting:

**Corollary 4.2.** *Theorem 4.1 may be generalized to multiple classes $\mathcal{Y} = \{1, 2, ..., m\}$ and multiple groups $\mathcal{G} \in \{1, 2, ..., n\}$, where $\beta_{g,y} = \Pr\left( \hat{Y}=y \mid G=g \right)$ and assuming $\mathrm{Var}_{\mathcal{S}}[\omega_g(\mathcal{T}, \mathcal{S}, x)] \leq B_g$:*

$$\Delta^\star_{\mathsf{DP}}(\pi, \mathcal{T}) \coloneqq \sum_{y \in \mathcal{Y}} \sum_{g, h \in \mathcal{G}} \left| \Pr_{\pi, \mathcal{T}}(\hat{Y}=y \mid G=g) - \Pr_{\pi, \mathcal{T}}(\hat{Y}=y \mid G=h) \right| \tag{14}$$

$$\Delta^\star_{\mathsf{DP}}(\pi, \mathcal{T}) \leq \Delta^\star_{\mathsf{DP}}(\pi, \mathcal{S}) + \sum_y \sum_g \left( \beta_{g,y}(1 - \beta_{g,y}) \cdot B_g \right)^{1/2} \tag{15}$$

We remark that in general, binary classification bounds may frequently be generalized to multi-class bounds by redefining fairness violations as a sum of binary-class fairness violations (*i.e.*, same-class *vs.* different-class labels) and summing the bounds on each.

### 4.2 Equal Opportunity

Consider an example using the $(Y=1)$-conditioned case of Equalized Odds—termed *Equal Opportunity* (EOp). Denoting, for each group $g$, the true positive rate $\beta_g^+ \coloneqq \Pr_{\pi, \mathcal{T}}(\hat{Y}=1 \mid Y=1, G=g)$ as an implicit function of $\pi$ and $\mathcal{T}$, we define disparity for EOp as $\Delta^\star_{\mathsf{EOp}}(\pi, \mathcal{T}) \coloneqq \sum_{g, h \in \mathcal{G}} |\beta_g^+ - \beta_h^+|$.

We may bound the realized value of $\Delta^\star_{\mathsf{EOp}}(\pi, \mathcal{T})$ by bounding $\beta_g^+$ for each group:

**Theorem 4.3.** *Subject to covariate shift and any given $\mathbf{D}, \mathbf{B}$, assume extremal values for $\beta_g^+$, i.e.,*

$$\forall g, \ \left( D_g(\mathcal{T} \parallel \mathcal{S}) < B_g \right) \implies \left( l_g \leq \beta_g^+(\pi, \mathcal{T}) \leq u_g \right) \tag{16}$$

*it follows that*

$$v(\Delta^\star_{EOp}, \mathbf{D}, \pi, \mathcal{S}, \mathbf{B}) \leq \max_{\substack{x_g \in \{l_g, u_g\} \\ x_h \in \{l_h, u_h\}}} \sum_{g,h} \left| x_g - x_h \right| \tag{17}$$

**Corollary 4.4.** *The disparity measurement $\Delta^\star_{EOp}$ cannot exceed $\frac{|\mathcal{G}|^2}{4}$.*

In Appendix D, we bound the extremal values of $\beta_g^+$ by geometrically interpreting this quantity as an inner product on an appropriate vector space, utilizing the freedom to select an appropriate $\mathbf{D}$.

# 5 Label Shift

Under label shift ($\Pr_\mathcal{S}(X|Y) = \Pr_\mathcal{T}(X|Y)$), violations of EO and EOp are invariant, because the independence of $\hat{Y}$ and $Y$ given $X$ implies $\Pr_{\pi,\mathcal{T}}(\hat{Y}|Y) = \Pr_{\pi,\mathcal{S}}(\hat{Y}|Y)$. We therefore focus on the violation of demographic parity (DP) (Equation (3)) subject to the label shift condition, treating a binary classification task over two groups for simplicity.

In this setting, we choose to measure group-specific distribution shifts from $\mathcal{S}$ to $\mathcal{T}$ by the change in proportion of ground-truth positive labels, which we refer to as the group *qualification rate* $Q_g(\mathcal{T}) := \Pr_\mathcal{T}(Y = 1 \mid G = g)$:

$$D_g(\mathcal{T} \parallel \mathcal{S}) := \left| Q_g(\mathcal{S}) - Q_g(\mathcal{T}) \right| \leq B_g \tag{18}$$

**Theorem 5.1.** *A Lipshitz condition bounds $\nabla_\mathbf{b} v(\Delta^\star_{DP}, \mathbf{D}, \pi, \mathcal{S}, \mathbf{b})$ when*

$$D_g(\mathcal{T} \parallel \mathcal{S}) := \left| Q_g(\mathcal{S}) - Q_g(\mathcal{T}) \right| \leq B_g \tag{19}$$

*Specifically,*

$$\frac{\partial}{\partial b_g} v(\Delta^\star_{DP}, \mathbf{D}, \pi, \mathcal{S}, \mathbf{b}) \leq (|\mathcal{G}| - 1) \left| \beta_g^+ - \beta_g^- \right| \tag{20}$$

*for true positive rates $\beta_g^+$ and false positive rates $\beta_g^-$:*

$$\beta_g^+ := \Pr_\pi(\hat{Y}{=}1 \mid Y{=}1, G{=}g); \quad \beta_g^- := \Pr_\pi(\hat{Y}{=}1 \mid Y{=}0, G{=}g) \tag{21}$$

Because $\beta_g^+$ and $\beta_g^-$ are invariant under label shift given a constant policy $\pi$, we elide their explicit dependence on the underlying distribution.

**Theorem 5.2.** *For DP under the bounded label-shift assumption $\forall g, |Q_g(\mathcal{S}) - Q_g(\mathcal{T})| \leq B_g$,*

$$\Delta^\star_{DP}(\pi, \mathcal{T}) \leq \Delta^\star_{DP}(\pi, \mathcal{S}) + (|\mathcal{G}| - 1) \sum_g B_g \left| \beta_g^+ - \beta_g^- \right| \tag{22}$$

Intuitively, the change in $\Delta^\star_{DP}$ subject to label shift depends on $|\beta_g^+ - \beta_g^-|$, the marginal change in acceptance rates as agents change their qualifications $Y$. We measure the distribution shift as agents change their qualifications by $|Q_g(\mathcal{S}) - Q_g(\mathcal{T})|$. When $\beta_g^+$ is close to $\beta_g^-$, the policy looks like a random classifier, and a label shift has limited effect on statistical group disparity. When $|\beta_g^+ - \beta_g^-|$ is large, indicating high classifier accuracy, the effect on supremal disparity is larger. Our bound thus exposes a direct trade-off between accuracy and fairness transferability guarantees.

# 6 Comparisons to Synthetic Distribution Shifts (Demographic Parity)

To further interpret our results, in this section, we consider specific and popular agent models to characterize distribution shift and instantiate our bounds for particular forms of $\mathbf{D}$, $\mathbf{B}$, and $\Delta^\star$.

## 6.1 Covariate Shift via Strategic Response

Let us consider a specific example of covariate shift (Equation (6)) caused by a deterministic, group-independent model of *strategic response* in which agents react to a binary classification policy $\pi$ characterized by group-specific feature thresholds:

$$\hat{Y} \sim \pi(x, g) = \begin{cases} 1 & \text{with probability 1 if } x \geq \tau_g \\ 0 & \text{with probability 1 otherwise} \end{cases} \tag{23}$$

For simplicity, we assume the feature domain $\mathcal{X} = [0,1]$. In response to threshold $\tau_g$, agents in each group $g$ may modify their feature $x$ to $x'$ by incurring a cost $c_g(x, x') \geq 0$. Similar to [18], we define the *utility* $u_g$ for agents in group $g$ to be

$$u_g(x, x') := \beta_g(x') - \beta_g(x) - c_g(x, x'); \quad \beta_g(x) := \Pr\left(\hat{Y}=1 \mid X=x, G=g\right), \forall g. \tag{24}$$

Contrary to the standard strategic classification setting, we do not assume that feature updates represent false reports, but that such updates may correspond to actual changes underlying the true qualification $Y$ of each agent. This assumption has been made in a recent line of research in incentivizing improvement from human agents subject to such classification [5].

Next, we assume all agents are rational utility maximizers (Equation (24)). For a given threshold $\tau_g$ and manipulation budget $m_g$, the best response of an agent with original feature $x$ is

$$x' = \underset{z}{\arg\max}\ u_g(x, z), \quad \text{such that}\ c_g(x, z) \leq m_g \tag{25}$$

To make the problem tractable, we make additional assumptions about the agents' best responses.

**Assumption 6.1.** An agent's original feature $x$ is sampled as $X \sim \mathcal{U}_{[0,1]}$[4].

**Assumption 6.2.** The cost function $c_g(x, x')$ is monotone in $|x - x'|$ as $c_g(x, x') = |x' - x|$.

Under Assumption 6.2, only those agents with features $x \in [\tau_g - m_g, \tau_g)$ will *attempt* to change their feature. We also assume that feature updates are non-deterministic, such that agents with features closer to the decision boundary $\tau_g$ have a greater *chance* of updating their feature and each updated feature $x'$ is sampled from a uniform distribution depending on $\tau_g$, $m_g$, and $x$:

**Assumption 6.3.** For agents who *attempt* to update their features, the probability of a successful feature update is $\Pr(X \neq X') = 1 - \frac{|x - \tau_g|}{m_g}$.

**Assumption 6.4.** An agent's updated feature $x'$, given original feature $x$, manipulation budget $m_g$, and classification boundary $\tau_g$, is sampled as $X' \sim \mathcal{U}_{[\tau_g, \tau_g + m_g - x]}$.

With the above setting, we can specify the reweighting coefficient $\omega_g(x)$ for our setting (Equation (102) in Appendix F.1) and get the following bound for the strategic response setting[5]:

**Proposition 6.5.** *For our assumed setting of strategic response involving* DP *for two groups $\{g, h\}$, Theorem 4.1 implies*

$$\Delta_{DP}^{\star}(\pi, \mathcal{T}) \leq \Delta_{DP}^{\star}(\pi, \mathcal{S}) + \tau_g(1 - \tau_g)\frac{2}{3}m_g + \tau_h(1 - \tau_h)\frac{2}{3}m_h \tag{26}$$

The above result shows that two factors lead to a smaller difference between the source and target fairness violations: a less stochastic classifier (when the threshold $\tau_g$ is far away from $0.5$) and a smaller manipulation budget $m_g$ (diminishing agents' ability to adapt their feature). In this case, $B_g = \frac{2}{3}\tau_g(1 - \tau_g)$. These factors lead to less potential manipulation and result in a tighter upper bound for the fairness violation on $\mathcal{T}$.

## 6.2 Label Shift via Replicator Dynamics

We now evaluate our theoretical bound for demographic parity subject to label shift (Theorem 5.2) on the replicator dynamics model of Raab and Liu [30]. Briefly, replicator dynamics assumes that the proportion of agents in a population choosing one strategy over another grows in proportion to the ratio of average utilities realized by the two strategies. The cited model additionally assumes $\mathcal{X} = \mathbf{R}$, $\mathcal{Y} = \{0,1\}$, and a monotonicity condition for $\mathcal{S}$ given by $\frac{d}{dx}\frac{\Pr_{\mathcal{S}}(X=x|Y=1)}{\Pr_{\mathcal{S}}(X=x|Y=0)} > 0$.

Label shift under the discrete-time ($t$) replicator dynamics may be expressed in terms of group *qualification rates* $Q_g := \Pr_t(Y=1 \mid G=g)$ and agent utilities (*i.e.*, group- and feature-independent values $U_{y,\hat{y}}$) such that, in each group, the popularity and average utility associated with a label determines its frequency at the next time step $t+1$.

---

[4]where $\mathcal{U}$ represents the uniform distribution.
[5]See Figure 5 in Appendix C for a demonstration of Theorem 4.1.

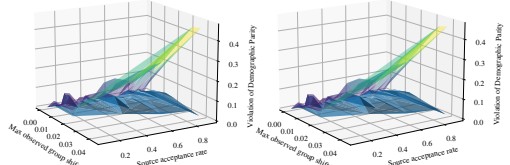

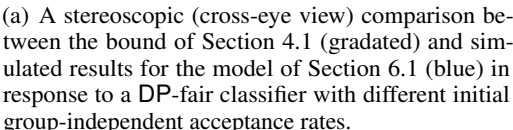

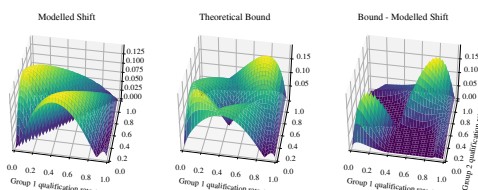

(a) A stereoscopic (cross-eye view) comparison between the bound of Section 4.1 (gradated) and simulated results for the model of Section 6.1 (blue) in response to a DP-fair classifier with different initial group-independent acceptance rates.

(b) A policy satisfying DP is subject to distribution shift prescribed by replicator dynamics (Section 6.2). Realized disparity increases (blue) are compared to the theoretical bound (Theorem 5.2, gradated), which is tight when group have dissimilar qualification rates.

**Figure 3:** Comparisons to synthetic distribution. Larger versions are provided in Appendix E.

Denote the *fractions* of group-conditioned, feature-independent outcomes with the expression $\rho_g^{y,\hat{y}} :=$ $\Pr_t(\hat{Y}=\hat{y}, Y=y \mid G=g)$ and abbreviate the fraction-weighted utility as $u_g^{y,\hat{y}}(t) := U_{y,\hat{y}} \cdot \rho_g^{y,\hat{y}}$. We may then represent the replicator dynamics as

$$Q_g[t+1] = \frac{u_g^{1,1}(t) + u_g^{1,0}(t)}{u_g^{1,1}(t) + u_g^{1,0}(t) + u_g^{0,0}(t) + u_g^{0,1}(t)} \tag{27}$$

To apply Theorem 5.2, we also observe that $|\beta_g^+ - \beta_g^-| = \frac{|\rho_g^{1,1} - \rho_g^{0,1}|}{\rho_g^{1,1} + \rho_g^{0,1}}$, where $\beta_g^+$ and $\beta_g^-$ represent the true positive rate and false positive rate for group $g$, respectively, and we use the change in qualification rate as our measurement of label shift, *i.e.*, $B_g = |Q_g[t+1] - Q_g[t]|$. When demographic parity is perfectly satisfied, we note that the acceptance rate ($\rho_g^{1,1} + \rho_g^{0,1}$) is group-independent.

**Theorem 6.6.** *For DP subject to label replicator dynamics,*

$$\Delta_{DP}^\star(\pi, \mathcal{T}) \leq \Delta_{DP}^\star(\pi, \mathcal{S}) + \sum_g \left| Q_g[t+1] - Q_g[t] \right| \frac{|\rho_g^{1,1} - \rho_g^{0,1}|}{\rho_g^{1,1} + \rho_g^{0,1}} \tag{28}$$

In Figure 3(b), we graphically represent all possible states of an initially fair system (thus determining $\beta$ and $\rho$ as a result of the monotonicity condition) by the tuple of qualification rates for each group. With the dynamics prescribed by Equation (27), we depict the *rate of change* of disparity given a fixed, locally DP-fair policy, and compare this to the theoretical bound when $B_g = |Q_g[t+1] - Q_g[t]|$.

Interpreting our results, we note that the bound lacks information about the relative directions of the change in acceptance rates for each group, and thus over-approximates possible fairness violations when group acceptance rates shift the same direction. When group acceptance rates move in opposing directions, however, the bound gives excellent agreement with the modelled replicator dynamics.

## 7 Comparisons to Real-World Distribution Shifts

We now compare our special-case theoretical bounds (*i.e.*, label/covariate shift) to real-world distribution shifts and hypothetical classifiers. We use American Community Survey (ACS) data provided by the US Census Bureau [16]. We adopt the sampling and pre-processing approaches following the Folktables package provided by Ding et al. [13][6] to obtain 1,599,229 data points. The data is partitioned by (1) all fifty US states and (2) years from 2014 to 2018. We use 10 features covering the demographic information used in the UCI Adult dataset [4], including age, occupation, education, *etc.*, as $X$ for our model, select sex as binary protected group, i.e., $G \in \{g = \texttt{female}, h = \texttt{male}\}$. We set the label $Y$ to whether an individual's annual income is greater than \$50K.

To apply our label-shift or covariate-shift bounds, we first need to verify whether the two datasets satisfy either of these assumptions. We adopted a conditional independence test [22], which takes data from source and target domains as input and returns a divergence score for each covariate and label variable, reflecting to what extent the variable is shifted between distributions. We find that the likelihood that the covariates shift across US states is approximately two orders of magnitude higher

---

[6]This package is available at `https://github.com/zykls/folktables`.

than for labels. More specifically, there are 4 covariates, including class of worker (probabilistic divergence score of 2.67e-2), hours worker per week (3.56e-2), sex (3.56e-2) and race (2.55e-1), that are more likely to be shifted than the label variable (1.29e-4). For temporal shifts within states, we find that the label variable is more likely to be shifted (0.1) than all the other covariates (which are below 0.01), approximately two orders of magnitude in favor of label shift over covariate shift. We therefore compare the disparities of hypothetical policies on these distributions to bounds generated from the corresponding, approximately satisfied assumptions.

On this data, we train a set of group-dependent, linear threshold classifiers $\mathrm{Pr}_{\pi(x,g)}(\hat{Y}=1) = \mathbb{1}[\sigma(w \cdot x) > \tau_g]$, for a range of thresholds $\tau_g$ and $\tau_h$ for each source distribution. Here, $\sigma(\cdot)$ is the logistic function and $w$ denotes a weight vector. We then consider two types of real-world distribution shift: (1) *geographic*, in which a model trained for one state is evaluated on other US state in the same year, and (2) *temporal*, in which a model trained for 2014 is evaluated on the same state in 2018.

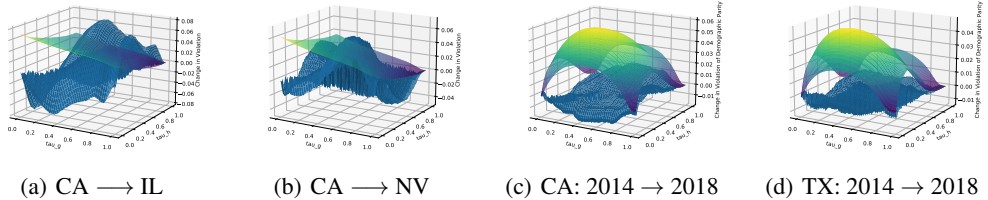

| (a) CA ⟶ IL | (b) CA ⟶ NV | (c) CA: 2014 → 2018 | (d) TX: 2014 → 2018 |

**Figure 4:** Simulated change in DP violation (blue mesh) subject to geographic and temporal distribution shifts *vs.* direct application of bounds for approximately satisfied assumptions (respectively, Theorem 4.1 and Theorem 5.2) (gradated mesh). The $x$-axis and $y$-axis of both figures represent the policy thresholds $\tau_g$ and $\tau_h$.

We graphically compare the theoretical bounds of Theorem 4.1 and Theorem 5.2 for the increased violation of DP subject to covariate and label shift, respectively, to the simulated violations for our model and data in Figure 4. We provide additional examples and an evaluation of bounds for EO subject to covariate shift (noting that label shift preserves EO in theory) in Appendix E.2. Despite the fact that geographic or temporal distribution shifts only approximately satisfy the assumptions of covariate or label shift, these comparisons demonstrate that our theoretical bounds are not vacuous, approximately bounding the change of fairness violation across real-world domain shifts. For geographic shifts, the covariate shift EO bounds (Appendix E.2) correctly overestimate disparity and tighten near accurate policies, while our DP bounds are useful only for a subset of policy thresholds (Figure 4(b)). add specific pointer. e.g, 4.a, that one is 4.b For temporal shift, the label shift bound for DP correctly overestimates the real change of DP violations but still remains at the same order of magnitude (Figure 4(c) and 4(d)).

## 8 Conclusion and Discussion

In this paper, we have developed a unifying framework for bounding the violation of statistical group fairness guarantees when the underlying distribution shifts within presupposed bounds. We hope that this work can generate meaningful discussion regarding the viability of fairness guarantees subject to distribution shift, the bounds of adversarial attacks against algorithmic fairness, and evaluations of robustness with respect to algorithmic fairness. We believe that, just as published empirical measurements are of limited use without reported uncertainties, fairness guarantees must be accompanied by bounds on their robustness to distribution shift.

Future work remains to apply our framework for to problem of fairness transferability in settings with more complicated distribution shift dynamics. For example, compound distribution shifts [33], which compose covariate shifts and label shifts, cannot be treated by composing the theoretical bounds developed herein without additional information regarding intermediate distributions. Another potential future direction is to develop reasonable bounds on anticipated distribution shift from models of human behavior and exogenous pressures.

**Acknowledgement**  This work is supported by the National Science Foundation (NSF) under grants IIS-2143895, IIS-2040800 (FAI program in collaboration with Amazon), and CCF-2023495.

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
