# OpenReview forum: "Fairness Transferability Subject to Bounded Distribution Shift"
_NeurIPS.cc/2022/Conference — NeurIPS 2022 Accept_

### Official Review · Reviewer_ifHr · 2022-07-04

**Rating:** 5
**Confidence:** 3
**Soundness:** 3 good
**Presentation:** 2 fair
**Contribution:** 3 good

**Summary:**

This paper studies the transferability of group fairness under distribution shift which is an important problem in reality. Under bounded distribution shift, this paper develops a general bound for the fairness violation in the test time. It also derives bounds when considering particular fairness metrics (including demographic parity and equalized odds) and types of distribution shift (including covariate shift and label shift).


**Questions:**

1. The theoretical bounds depend on the source and target data distribution, but in practice, we only have finite samples. What’s the estimation error when we measure the bound in practice? Are there any factors, such as the data size, that affect the measurement in practice?
2. Based on the theoretical bounds, do you have any ideas or suggestions for the algorithms to tackle the distribution shift? How to design policy \pi so that to have a smaller upper bound?
3. Does figure 4 indicates that the actual distribution shift is not a covariate shift, and the bounds become less useful in practice?

Minor issue:
In equation (22), is there a typo under $\max$? Do you miss $x_h$?


**Limitations:**

One limitation of this paper is that the theoretical bounds are based on some impractical assumptions which hinder their use in reality. I encourage authors to discuss some potential algorithms inspired by the theoretical bounds.

**Strengths And Weaknesses:**

Strengths:
This is the first paper that provides guarantees of fairness under bounded distribution shift. This paper considers multiple types of distribution shift and multiple fairness metrics. They also compare the derived theoretical bounds with the real change in fairness violation. The theoretical analysis can be applied to any models.

Weakness:
1. This paper has too many notations. I got lost many times.
2. It would be better if there are more intuitions and explanations. For example, what’s the message we can get from theorem 3.2? Figure 2 is hard to understand without explanation.
3. Many contents are delayed to the appendix, but the appendix is not included in the supplementary material. (Please point it out if I missed it.)
4. I’m concerned about the practical use of the derived bounds. This is because the assumptions this paper considers including the particular distribution shift (covariate shift or label shift) and the availability of the target data are hard to be satisfied in reality.

---

> ### Author Response · Authors · 2022-08-02
> **Response to Reviewer ifHr**
>
> Thank you for the feedback. Below we address your comments and questions:
>
> > Too many notations
>
> We apologize for having heavy notations in our paper. As we mentioned in the general response, the appendices (now included in the supplementary material) feature a table (see Table 1 in Section A in the appendix) of notation that we believe will benefit future readers.
>
> > It would be better if there are more intuitions and explanations. For example, what’s the message we can get from Theorem 3.2? Figure 2 is hard to understand without explanation.
>
> The intuition behind Theorem 3.2 is this: If we are guaranteed that disparity can never increase faster than a certain rate in some measure of distribution shift, then, given a maximum distribution shift, this rate bounds the maximum possible disparity. We have included this language in our revised paper.
>
> Also in revision, Figure 2 has been made clearer by clarifying that $\max \partial_g v = l_g$. That is, the maximal components of slope of the surface $v$ in the figure correspond to the components of $L$. The red dotted line in the figure corresponds to $l_h b_h$ while the blue dotted line corresponds to $l_g b_g$. Adding these quantities results in the dot product $L \cdot B$.
>
> > Many contents are delayed to the appendix, but the appendix is not included in the supplementary material. (Please point it out if I missed it.)
>
> The appendix has been attached in revision. We apologize for omitting it, in error, upon initial submission.
>
> > I’m concerned about the practical use of the derived bounds…. because … the assumptions [about particular distribution shifts] ... and the availability of the target data.
>
> We address this concern about distribution shift assumptions in the general rebuttal under the heading “2. Practical Use”. With respect to target distribution data, our bounds do not require data from the target distribution. Instead, they require a given limit on the maximum measure of distribution shift that may occur.
>
> > The theoretical bounds depend on the source and target data distribution, but in practice, we only have finite samples. What’s the estimation error when we measure the bound in practice? Are there any factors, such as the data size, that affect the measurement in practice?
>
> One way to view the role of sampling in our framework is to add the amount of distribution shift between the sampled (training) distribution and the “true” source distribution (as a sample complexity bound) to the maximum anticipated distribution shift between the source and target distributions, prior to determining maximum realizable disparity.
>
> Ultimately our theoretical bounds ultimately treat $\mathcal{S}$ as the *empirical* source distribution (on which the policy is trained), not the “true” source distribution (from which data is sampled). In this way, divergence between the empirical and true source distribution caused by finite sample sizes serve as a use-case for our bounds: That is, the number of samples used in training provide probabilistic bounds on how “far away” the empirical distribution (for which the policy is derived) may be from the true source distribution (on which it may be used). Within our framework, this error translates to bounds on how unfair the policy may be on the true source distribution during deployment.
>
> For relevant work on translating finite samples to distribution shifts, we reference two papers in Appendix B, within the paragraph on “Domain Adaptation” (We apologize for omitting the appendix in our original submission, in error).
>
> > Based on the theoretical bounds, do you have any ideas or suggestions for the algorithms to tackle the distribution shift? How to design policy \pi so that to have a smaller upper bound?
>
> In general, our bounds are not prescriptive for choosing policies, merely assigning adversarial robustness guarantees to fairness measures. We address, in our general rebuttal under heading "#2 Practical Use", how robustness may be treated as a cost in general optimization methods for choosing policies.
>
> > Does figure 4 indicates that the actual distribution shift is not a covariate shift, and the bounds become less useful in practice?
>
> We address this concern in our general rebuttal under the heading “3. Interpretation of Section 7”. While the bounds are no longer strict bounds when their underlying assumptions are violated, they still retain utility, and future work promises futher bounds with more appropriate assumptions.
>
> > Minor issue: equation (22), is there a typo
>
> Yes. Thank you for pointing out this error. This has been fixed in revision.

---

> ### Author Response · Authors · 2022-08-07
> **Could you kindly check whether our response and updated paper properly addressed your concern?**
>
> Dear Reviewer ifHr,
>
> Once again, thank you very much for reviewing this paper. We have provided responses to your comments and revised our submission. Could you please let us know if we have properly addressed your concerns? If there are any additional concerns, we hope we will have the opportunity to respond to them. Your feedback is appreciated!
>
> Best,
> The Authors

---

### Official Review · Reviewer_fUG9 · 2022-07-11

**Rating:** 8
**Confidence:** 3
**Soundness:** 3 good
**Presentation:** 2 fair
**Contribution:** 4 excellent

**Summary:**

This paper's primary contribution is a theoretical upper bound on certain fairness definitions, when the target distribution has drifted from the source distribution. The paper then works through how label shift and covariate shift would affect the bounds on certain fairness definitions, leading to a brief empirical analysis of the results. The following are the main contributions:

1) The paper provides a theoretical result that shows an upper bound on the violations distribution shifts can achieve under several different fairness definitions. (Significance: High)

2) The paper provides a walk through of how certain distribution shift characteristics can be analyzed with their framework. (Significance: Medium)

3) The paper provides a unifying framework for several other related works trying to assess the impacts of distribution shift (Significance: High)

**Questions:**

1) Could you give a more granular description of Appendix E.2. and why it was relegated to the appendix instead of being written in section 7?

**Limitations:**

As this paper is theoretical, there doesn't seem to be any limitations to discuss. If anything, the paper provides upper bounds on the limitations of other papers that explore how distribution shifts affect the violation of fairness definitions.

**Strengths And Weaknesses:**

Quality:

The paper is of high quality overall, but section 7 needs a bit more work. The paper suggests in the abstract that empirical justification would be given for both demographic parity and equalized odds. However, section 7 seems to only deal with the demographic parity definition without much discussion. Recommendations:

1) Clarify the abstract as to what empirical justification will be given

2) Bolster section 7 with either examples concerning equalized odds and/or give more discussion as to how to interpret the outputs of the empirical justification.

Possibly, including details from Appendix E.2. in section 7 may solve this issue.

Clarity:

Overall, the paper is clear, but the over reliance on notation makes it hard to read at times. Many definitions have overused notation (such as implication symbols) rather than plain English, making interpretation of the definition hard to gauge. As an example, please look at definition 2.4, where the if and only if symbol and plain English are both there.

Originality:

Based on my knowledge of the field and the authors’ citations, I believe that this is original work that will move the field forward. However, I do find that some critical dynamics in fairness papers are missing from the citations, such as Liu et. al. 2018 Delayed Impact of Fair Machine Learning and Creager et. al. 2020 Causal Modeling for Fairness in Dynamical Systems, which takes papers like Liu et. al. 2018 and simplifies them into a single framework as well. These citations will provide more credibility and offer other avenues for unifying frameworks for dynamics in fairness.

Significance:

This paper provides a significant result for the field that could provide a general bound for how much dynamics can violate fairness definitions on a static dataset. Although I am unsure right now how to use it in practice, I believe having these theoretical tools will be helpful in the future. However, the paper could be made more legible by decreasing the usage of notation. Moreover, having a more robust narrative in section 7 will show the results in the real world better.

---

> ### Author Response · Authors · 2022-08-02
> **Response to Reviewer fUG9**
>
> Thank you for the feedback. Below we address your comments and questions:
>
> > Section 7 seems to only deal with the demographic parity definition without much discussion.
>
> We have included an extended discussion of Section 7 in our general rebuttal under the heading of “3. Interpretation of Section 7” and have revised our paper to justify the use of these special-case bounds for real-world distribution shifts and interpret the results.
>
> > Bolster section 7 with either examples concerning equalized odds and/or give more discussion as to how to interpret the outputs of the empirical justification.
>
> Also mentioned under heading “3. Interpretation of Section 7” of our general rebuttal, we have also included a bound for Equalized Opportunity (EO) subject to covariate shift in the appendix, but note that no such bound exists for label shift, which preserves EO (see first paragraph of Section 5)
>
> > Could you give a more granular description of Appendix E.2. and why it was relegated to the appendix instead of being written in section 7?
>
> We apologize for neglecting to attach the Appendices in our initial submission. Our revision includes more discussion within Section 7 and restores the Appendices. Appendix E.2 provides graphics for additional state to state and temporal distribution shifts.
>
> > Clarify the abstract as to what empirical justification will be given.
>
> We include a sentence about our empirical results in the abstract.
>
> > Dynamics in fairness papers are missing from the citations.
>
> Thank you for pointing out the additional citations on dynamics in fairness. As we mentioned in the general response, we have added these citations to the revised version of our paper with additional discussions in the Appendix.

---

> ### Author Response · Authors · 2022-08-07
> **Could you kindly check whether our response and updated paper properly addressed your concern?**
>
> Dear Reviewer fUG9,
>
> Once again, thank you very much for reviewing this paper. We have provided responses to your comments and revised our submission. Could you please let us know if we have properly addressed your concerns? If there are any additional concerns, we hope we will have the opportunity to respond to them. Your feedback is appreciated!
>
> Best,
> The Authors

---

> > ### Comment · Reviewer_fUG9 · 2022-08-08
> > **Appropriately addresses my concerns**
> >
> > Thank you for your rebuttal and associated edits. They address my concerns, and then some. The fact that geographic and temporal shifts only approximately satisfy the assumptions of label and covariate shift make sense, as the first image (CA -> IL) isn't a complete upper bound. I will adjust my rating accordingly.

---

> > > ### Author Response · Authors · 2022-08-08
> > > **Thank you, reviewer fUG9!**
> > >
> > > Thank you again for your review and your kind reevaluation following our rebuttal. We greatly appreciate your time and assistance!

---

### Official Review · Reviewer_84wj · 2022-07-11

**Rating:** 5
**Confidence:** 3
**Soundness:** 3 good
**Presentation:** 3 good
**Contribution:** 2 fair

**Summary:**

This paper considers fairness under domain shift, specifically, bounding the change in the violation of group fairness criteria when the evaluation (target) data distribution differs from the one (source) on which the model is trained.

- First, a framework/distribution shift assumption is proposed for studying this problem.  Namely, the domain shift should be bounded w.r.t. some probability (pre)metric.  Then the worse-case (or adversarial) fairness bound is considered under this assumption.
- The authors derive some results under several simplifying assumptions in section 3, then work out the results for covariate shift and label shift in sections 4 and 5, respectively.
- In section 6, two examples of distribution shifts are considered, which are instances of covariate and label shift.
- Finally in section 7, a visualization of the bounds in sections 4 and 5 are provided on a real-world dataset, and are compared to the true change in fairness violation incurred by a model under simulated domain shift.

**Questions:**

My main questions are embedded in the section above.

1. line 230 missing $x_h\in\{l_h,u_h\}$ under $\max$?


**Limitations:**

The problem studied is of interest to the field, but the potential impact of the present work may be limited given the assumptions required in the results.  I am willing to hear the opinions of the other reviewers.


**Strengths And Weaknesses:**

Strengths.

1. The paper is complete, clearly written, and easy to follow.
2. The problem studied, fairness under domain shift, is current and of importance to the field.

Weaknesses.

1. I feel that some of the results are weak.

    - It is unclear whether the results in section 3 under the simplifying assumptions would be generally useful; as the authors pointed out on line 183 “direct computation of [terms in theorem 3.2] is difficult”, and the same could be said of theorem 3.4.

        Although the proof of theorem 5.2 uses theorem 3.2, I feel that a direct proof with potentially simpler arguments is also possible, given the label shift assumption.

    - I feel that the conclusion in theorem 4.3 is trivial given the assumption that the change in true positive rate is bounded.  Evaluating the change in TPR requires labels from the target distribution; is there a bound if this information is not available?

2. It is not clear what is being visualized in fig. 4 in section 7.  Is the blue mesh the ground-truth change in DP, and the colormapped mesh the modeled change?  Are the left two plots produced assuming covariate shift, and the right two label shift?  Are there any remarks/comments on the results?

---

> ### Author Response · Authors · 2022-08-02
> **Response to Reviewer 84wj**
>
> Thank you for the feedback. Below we address your comments and questions:
>
> > General usefulness of results in Section 3 … (“direct computation of [terms in theorem 3.2] is difficult”)
>
> This appears to be a misunderstanding that we address in our general rebuttal under heading “2. Practical Use”. In short, the potentially complex relationship between disparity and distribution shift may be simplified when considering only worst-case scenarios, and our framework may be used to tractably bound otherwise difficult-to-compute quantities.
>
> The text highlights that the benefit of Theorem 3.2 occurs when direct computation of $v$ is difficult, because Theorem 3.2 does not require its direct computation, merely a Lipshitz bound on its rate of change. Likewise, Theorem 3.4 is useful in that it relies only on local information about the structure of $v$ (i.e., the current measure of disparity and its first derivative with respect to distribution shift), rather than global information about its behavior (i.e., the actual functional form of disparity vs. distribution shift).
>
> > Direct proof for Theorem 5.2 (Although the proof of theorem 5.2 uses theorem 3.2, I feel that a direct proof with potentially simpler arguments is also possible, given the label shift assumption.)
>
> You are correct that Theorem 5.2 can be proved using simpler arguments – under label shift assumption, one can show that for each group $g$, $\mathsf{TPR}_g = \Pr (h(x) = +1 \mid Y = +1, G = g)$ and $\mathsf{FPR}_g = \Pr (h(x) = +1 \mid Y = -1, G = g)$ remain the same before and after the distribution shift. Using this, we can further express the positive prediction rate $(\Pr (h(x) = +1|G = g))$ as a weighted combination of $\mathsf{TPR}_g$ and $\mathsf{FPR}_g$, and then we can bound the difference between the positive prediction rates before and after the distribution shift explicitly using $\mathsf{TPR}_g$ and $\mathsf{FPR}_g$.
>
> > Theorem 4.3 (I feel that the conclusion in theorem 4.3 is trivial …)
>
> Theorem 4.3, while perhaps trivial in isolation, is an important stepping stone for Theorem D.1 (in Appendix D of the supplementary material), which provides a novel geometric (i.e., Hilbert Space) picture of distribution shift and which we use to bound the effect of covariate shift on Equal Opportunity.
>
> > Evaluating the change in TPR requires labels from the target distribution; is there a bound if this information is not available?
>
> Within our framework, a bound on the shifted value of TPR still does not require any data from the target distribution; it may result from assumptions about the dynamics. In general, we derive bounds that do not require labels from the target distribution, only the source distribution, because we treat all possible distribution shifts within certain bounds.
>
> > Visualization of Fig 4
>
> This is correct. The blue mesh represents the real change of demographic parity, and the colormapped mesh represents the bounds we predicted. Aligning with the caption of Figure 3, we will add text to explain this.  We further address Fig 4 and Section 7 in our general rebuttal, under heading “3. Interpretation of Section 7” and remark on the results.
>
> > Line 230 (equation 22) missing $x_h$
>
> Thank you for pointing out this error. We have corrected it in revision.

---

> ### Author Response · Authors · 2022-08-07
> **Could you kindly check whether our response and updated paper properly addressed your concern?**
>
> Dear Reviewer 84wj,
>
> Once again, thank you very much for reviewing this paper. We have provided responses to your comments and revised our submission. Could you please let us know if we have properly addressed your concerns? If there are any additional concerns, we hope we will have the opportunity to respond to them. Your feedback is appreciated!
>
> Best,
> The Authors

---

> ### Comment · Reviewer_84wj · 2022-08-08
> **Response to rebuttal**
>
> Thank you for the detailed responses to my comments, and I have increased my rating.  However, my doubts regarding the general usefulness of the results remain.
>
> Although the proposed framework and tools can be used to directly bound the change in DP violation under label/covariate shift, I feel that more direct and potential cleaner proofs are available if one only focuses on label/covariate shift; outside of these two special cases, I am uncertain of how the results may be applied generally.  For instance, in theorem 3.2, the Lipschitz condition $L$ could be interpreted as a bound on the change in DP violation as distribution shift increases, hence the $L\cdot B$ result follows.  But in general, I don't think it is easy to verify the Lipschitz condition.

---

> > ### Author Response · Authors · 2022-08-08
> > **Response to Reviewer 84wj's new response**
> >
> > Thank you again for your review and your kind reevaluation following our rebuttal. The comment on the practicality of Lipschitz condition is well-received. We agree this merits future works, but we want to provide some evidence that this can be done in practice. We’ll be happy to discuss further and clarify as needed.
> >
> >
> > > ...in general, I don't think it is easy to verify the Lipschitz condition
> >
> > You are correct that for a general fairness and distribution shift, it may be non-trivial to verify whether the fairness violation satisfies a Lipschitz condition.  When the proposed measures of disparity and distribution shift are also known to satisfy a concavity condition (i.e. the curvature of the changes of the topology of the fairness disparity is concave, see Fig 2 as a demonstration), however, a bounded local derivative of fairness disparity w.r.t. the measure of distribution shift will imply Lipschitzness. One may therefore empirically infer a Lipschitz condition from the source distribution by testing small, hypothetical distribution shifts and calculating the hypothetical resulting disparity (e.g., in analogy to using $(|(f(x+\delta)-f(x))/\delta|$ for a function $f$ around $x$). We leave a more sophisticated testing procedure for verifying a Lipschitzness condition as future work.
> >
> > Of course, it is equally valid to worry that concavity may not be typically encountered in the wild. When fairness measures and measures of distribution shift are not standard (e.g. compound distribution shift, or more general fairness constraints), we leave it to future research to apply our framework and exploit the structure of the setting. If concavity, Lipschitzness, or subadditivity hold, then the theorems already shown in our paper are directly applicable; many more generalizable results doubtless hold as well for other structural properties of the inherited measures. When measures are not fixed ahead of time, however, we may also choose them to have exploitable structural properties. In our paper, we do this by choosing a measure of distribution shift which is amenable to a geometric interpretation in the Appendix.
> >
> > As we believe that Lipschitz bounds are actually relatively common in the wild, we consider it important to include these results in Theorem 3.2, and we show that demographic parity subject to label shift is a common example that benefits from it (Section 5).

---

### Official Review · Reviewer_k7a3 · 2022-07-17

**Rating:** 8
**Confidence:** 4
**Soundness:** 3 good
**Presentation:** 3 good
**Contribution:** 3 good

**Summary:**

The paper studies the fair violations of predictive models under bounded distribution shifts. The main contributions are bounds on the change in fairness metrics (like demographic parity and equal opportunity) between train and test distributions that undergo either covariate shift or label shift. The bounds are further specialised to two applications in covariate or label shifts due to individuals strategically responding classifiers.

**Questions:**

Questions to address in the response

How can a practitioner use these to judge whether to use a model? Do the quantities included in the expressions (like Eq 17) plausibly known beforehand to estimate the max change in fairness?

How does the bounds or the analysis, in general relate to previous work? In what ways does the formulation unify those in the previous work, as claimed?

What are the takeaways from Adult Income results in Figure 4? Since the real data shifts do not neatly fit in either covariate or label shift, how to interpret the difference between the empirical and theoretical changes? I would suggest estimating the type of shifts and doing the comparison between states or time periods which have close to covariate or label shift. Or, to perform a more controlled setting where distribution shifts are simulated but the training data is real. Conditional independence test can be used to detect the types of shifts such as the methods used in this paper https://proceedings.neurips.cc/paper/2020/hash/e2d52448d36918c575fa79d88647ba66-Abstract.html

Relation of the results in Section 3.1, 3.2, and 3.3 to the results in Section 4 is not clear. Are the results in Section 4 special cases of these results?


Minor points and suggestions (I do not expect a response)

Some more related work
https://proceedings.neurips.cc/paper/2020/hash/d6539d3b57159babf6a72e106beb45bd-Abstract.html
https://arxiv.org/abs/2005.03474
https://proceedings.neurips.cc/paper/2021/hash/07563a3fe3bbe7e3ba84431ad9d055af-Abstract.html
https://arxiv.org/abs/2206.12796


Notation can be eased. For example, consider skipping \Psi or reusing \Delta for it since \Psi is not used in the results.

Please consider discussing how to bound the fairness change when both covariate and label distributions shift.

Describe the divergence metric D_g in Eq (21).

Figure 2 is not clear.

Describe the term replicator dynamics briefly.

Consider explaining why the divergence between distributions p and q is named ‘pre’metric in Definition 2.1.

**Limitations:**

Limitations of the work are not discussed. Please discuss limitations of the way distribution shifts are characterized that is the class of shifts considered and the parameters to quantify the shift, discuss the applicability of the results to help in deployment decisions, and the experiments required to validate the results on real data.

**Strengths And Weaknesses:**

Strengths
- Proposes a generic way to write down and analyze the maximum changes in fairness properties under shifts.
- Results are original. Analysis for three common fairness metrics and two broad types of shifts.
- The paper is well-written in general.

Weaknesses
- Significance of the bounds is not clear. The interpretation of the theoretical results does not describe how to apply them in practice.
- The experimental results on Adult Income data are not summarized.

The list of results seem to have non-obvious implications, but these are hard to see. Please address the questions below that will help me to decide whether to improve the evaluation to 6 or 7. Particularly, convey the insights from one of the bounds in more detail.

---

## After the response

The response aptly addressed my concerns on the practical utility and interpretation of the results. So I have increase my score to 8, Strong Accept.

The work makes a great contribution on anticipating robustness of fairness guarantees under distribution shift. It may encourage further work on making the shift assumptions more interpretable and practically useful.

---

> ### Author Response · Authors · 2022-08-02
> **Response to Reviewer k7a3 1/2**
>
> Thank you for the feedback. Below we address your comments and questions:
>
> > How can a practitioner use these [results] to judge whether to use a model? Do [certain] quantities [need to be known beforehand?]
>
> Yes, for deployment decisions, one requires the prospective policy, the source distribution, bounds on the possible realized distribution shift, and a maximum increase of disparity that will be tolerated. For example, in Theorem 4.1 (Eq 17), the measure of distribution shift (the variance of omega) must be bounded, while the acceptance rate $\beta$ is assumed to be known before deployment as determined by $\pi$ and the source distribution.
>
> > How does the bounds or the analysis, in general relate to previous work? In what ways does the formulation unify those in the previous work, as claimed?
>
> We address this question in our general response under heading “1. Relation to Prior Work”. In summary, recent studies have largely considered the transferability of specific measures of fairness subject to specific distribution shifts (e.g. label shift or covariate shift), assumptions, or dynamical models. We unify several measures of statistical group-fairness within a broader class of disparity functions (Section 2.2), and treat distribution shifts more generally, requiring only a means of measuring such shifts (Section 2.3).
>
> > What are the takeaways from Adult Income results in Figure 4? … how to interpret the difference between the empirical and theoretical changes?
>
> We address this question in our general response under heading “3. Interpretation of Section 7”. In short, Figure 4 demonstrates that the theoretical bounds are not vacuous, approximately bounding the change of fairness violation in the real world domain shifts. We clarify the use of color within the caption within the figures.
>
> > Use conditional independence test to detect the types of shifts
>
> We thank the reviewer for pointing out the use of conditional independence test to detect the types of shifts in our paper setting. We provide findings in the general response. In short, we find that the likelihood that the covariates shift between US states shift is approximately two orders of magnitude higher than labels, while for temporal shifts within states, approximately two orders of magnitude favor shifts in labels over covariates. We therefore compare these shifts to the bounds for their approximately satisfied assumptions. We add this to the revised version of our paper.
>
> > Relation of the results in Section 3.1, 3.2, and 3.3 to the results in Section 4 is not clear. Are the results in Section 4 special cases of these results?
>
> While some of our results in later sections may be proven with the results of Section 3, the results of Section 3 do not exhaust the types of bounds that can hold for our adversarial framework, nor must other possible bounds (Section 4, 5) derive from these results. In developing setting-specific bounds (combinations of label/covariate shift subject to demographic parity/equalized odds), we explore results independent of Section 3 to demonstrate the applicability of our framework on settings explored by previous research.
>
> > How to bound when both shift happens (Please consider discussing how to bound the fairness change when both covariate and label distributions shift.)
>
> In general, it is not possible to naively compose the bounds we derive for both covariate shift and label shift (i.e., subject to a combined “budget” for both types of distribution shift), because each bound independently requires knowledge of the distribution immediately prior to the represented shift. It follows that a composition of these two types of distribution shift requires, in addition, information about the specific intermediate distribution.
>
> Nonetheless, our framework is still applicable for general distribution shifts. Our initial results in this paper explore common types of bounds that may apply (Section 3), but we leave more specific examples of general distribution shifts to future work, opting to focus on familiar, restricted settings in this paper to showcase them as special cases. We discuss this issue further in our general rebuttal under heading “2. Practical Use”.
>
> > Describe the divergence metric D_g in Eq (21).
>
> Eq (21) describes a condition that must hold for any $D_g$ considered in Eq (22), but which is otherwise arbitrary.
>
> > Figure 2 is not clear.
>
> Figure 2 can be made clearer by explicitly noting that $\max \partial_g v = L_g$, that is, the maximal components of slope of the surface $v$ in the figure correspond to the components of $L$. The red dotted line in the figure corresponds to $L_h * b*h$ while the blue dotted line corresponds to $l_g * b_g$. Adding these quantities results in the dot product $L \cdot B$.
> We include this description in the revision to our paper.

---

> > ### Author Response · Authors · 2022-08-02
> > **Response to Reviewer k7a3 2/2**
> >
> > > Description of the term replicator dynamics.
> >
> > The replicator dynamics used in cited paper are distilled into Eq (31), but, briefly, replicator dynamics assumes that the proportion of agents in a population choosing between one strategy over another grows in proportion to the ratio of average utilities realized by the two strategies. We include this description in a revised version of our paper.
> >
> > > The naming of “pre”metric in Definition 2.1
> >
> > This terminology is borrowed the nomenclature from Wikipedia, despite a lack of consensus in the wider literature. We have added a footnote explaining what we mean by “premetric” for Definition 2.1. The name likely derives from the fact that it lacks some of the key axioms of a metric and is thus "prior" or "before" a full metric.
> >
> > > Limitations of the work are not discussed. Please discuss limitations of the way distribution shifts are characterized that is the class of shifts considered and the parameters to quantify the shift, discuss the applicability of the results to help in deployment decisions, and the experiments required to validate the results on real data.
> >
> > Thank you for these recommendations. We discuss these limitations in the revised manuscript accordingly. We clarify the applicability of the results in deployment decisions within Section 1.2, note limitations of our quantification of fairness in Section 2, and more completely discuss our experimental validation of results with real data in Section 7.

---

> ### Author Response · Authors · 2022-08-07
> **Could you kindly check whether our response and updated paper properly addressed your concern?**
>
> Dear Reviewer k7a3,
>
> Once again, thank you very much for reviewing this paper. We have provided responses to your comments and revised our submission. Could you please let us know if we have properly addressed your concerns? If there are any additional concerns, we hope we will have the opportunity to respond to them. Your feedback is appreciated!
>
> Best,
> The Authors

---

> ### Comment · Reviewer_k7a3 · 2022-08-07
> **After the response**
>
> Thanks for a thorough response. I appreciate the clarification on the real world shifts and interpretation of the results.
>
> The response aptly addressed my concerns on the practical utility and interpretation of the results. So I have increase my score to 8, Strong Accept. The work makes a great contribution on anticipating robustness of fairness guarantees under distribution shift. It may encourage further work on making the shift assumptions more interpretable and practically useful.

---

> > ### Author Response · Authors · 2022-08-07
> > **Thank you, Reviewer k7a3!**
> >
> > Thank you again for your review and your kind reevaluation following our rebuttal. We greatly appreciate your time and assistance!

---

### Author Response · Authors · 2022-08-02
**General Rebuttal, Part 1/3**

We thank our reviewers for the much appreciated feedback. Please allow us to address the salient, common concerns raised during the initial review here, followed by a summary of our revisions. We also respond to our reviewers individually with separate, official comments.

## 1. Relation to Prior Work (Reviewer k7a3 and fUG9)

Reviewers k7a3 and fUG9 both asked for us to provide additional details on how our work relates to existing literature, such as ensuring fair classification on the target distribution [1, 2, 3, 4], long term impact of fairness constraints [5], and causal modeling of fairness in a dynamical system [6]. Appendix B (in the revised supplementary material) largely addresses these concerns, but we also thank our reviewers for pointing out additional related work. We cite these additional papers in the revised manuscript.

Recent studies have largely considered the transferability of specific measures of fairness (e.g., equalized odds or demographic parity) subject to specific distribution shift assumptions (e.g. label shift or covariate shift) or dynamical models. We unify such measures of statistical group-fairness within a broader class of disparity functions (Section 2.2), with revised acknowledgement of the limitations of this formalism, and we treat distribution shifts more generally, requiring only a means of measuring per-group shifts (Section 2.3).

We see that the central question of fairness transferability has largely heretofore been “how should we design an algorithm such that it will be fair on a new (possibly known) target distribution?” Our work instead centers on the robustness of fairness guarantees for a given policy and source distribution, subject to distribution shifts. We ask, “given a classifier trained on an empirical, source distribution, what can we say about its fairness on all target distributions that differ within some bounds?”. To this end, we introduce an adversarial framework for judging fairness transferability. As we show in Section 3, this framework can produce general bounds without requiring specific distribution shift models or specific measures of fairness – merely structural properties between them. We also derive results for specific settings, explored by previous literature, as subcases within our framework — in Sections 4 and 5. Finally, in our revision, we outline types of distribution shifts that would be interesting for future work.

## 2. Practical Use (Reviewer k7a3, 84wj, ifHr):

Several reviewers questioned the practical application of our results.

To answer the first question posed by Reviewer k7a3, if a source distribution and prospective policy are given, along with bounds on maximum anticipated distribution shift, our framework provides a bound on the maximum violation of fairness that may be realized over all possible distribution shifts. By focusing on worst-case distribution shifts, we offer a tractable alternative to modeling potentially complex, real-world distribution shifts and dynamics. For the practitioner, we believe that such tractable bounds are necessary for certifying the robustness of a prospective policy’s fairness guarantees and for revealing potentially inappropriate deployments of machine learning, prior to deployment: That is, policies should not be deployed when potential disparities are greater than a selected threshold on some possible target distribution within a prespecified distribution shift of the training distribution.

To rebut a claimed weakness of our paper by Reviewer 84wj, in practice, it is often not necessary to calculate the worst-case target distributions nor worst-case disparities within our framework — a benefit when doing so would be computationally difficult. For example, in Section 3, we showcase bounds that require only structural properties between the measurements for fairness and distribution shift, rather than the computation of truly realizable disparities. In Theorem 3.2, our result depends only on (Lipshitz) smoothness conditions; in Theorem 3.4, a subadditivity condition guarantees that only local information is required to derive a bound. In both cases, we are spared from having to determine the explicit, worst-case distribution shift or disparities.

To rebut Reviewer ifHr’s 4th claimed point of weakness, we assert that it is also unnecessary to have any samples from the target distribution to utilize our bounds; our bounds do not require data from the target distribution. Instead, they require an upper limit on the maximum measure of distribution shift that may occur. In practice, we anticipate bonus on realizable distribution shifts to be produced by empirical modeling, dynamical assumptions, or regulatory policy.

To address Reviewer ifHr’s second question: while our framework is not prescriptive for finding policies with robust fairness guarantees, it may easily be incorporated into the constraints or cost terms of standard optimization procedures for finding such policies.

---

> ### Author Response · Authors · 2022-08-02
> **General Rebuttal, Part 2/3**
>
> ## 3. Interpretation of Section 7 (All Reviewers):
>
> All reviewers asked for additional clarification of the findings presented in Section 7 or Figure 4, in particular.
>
> It is a widely recognized problem in existing literature that real-world distribution shifts rarely conform precisely to standard models of distribution shift (e.g., label shift or covariate shift. See, e.g., [7]). In this regard, bounds based on such assumptions might prove inappropriate, however it is still valid to compare such bounds to real-world distribution shifts to determine, empirically, when such bounds approximately hold. For analogy, standard financial risk models may assume geometric brownian motion in underlying equity prices, because such models still yield useful predictions, despite evidence that these assumptions ultimately break down under close scrutiny.
>
> We have revised Section 7 and Figure 4 to better justify and explain the comparisons of such special-case bounds to empirical data. Specifically, we have adopted the conditional independence test suggested by Reviewer k7a3, which takes data from source and target domains as input and returns a divergence score for each covariate and label variable, reflecting to what extent the variable is shifted between distributions.
>
> For the Adult Income data, when comparing geographic states within the same year, there are 4 covariates, including class of worker (probabalistic divergence score of 2.67e-2), hours worker per week (3.56e-2), sex (3.56e-2) and race (2.55e-1), that are more likely to be shifted than the label variable (1.29e-4). Succinctly, the covariates are two orders of magnitude more likely to have shifted than the label variable, and we treat this shift (between geographic states in the same year), approximately, as a covariate shift for comparison to our bounds. Similarly, across years within the same state, we find that the label variable is more likely to be shifted (0.1) than all the other covariates (which are below 0.01). As this difference is again at least two orders of magnitude, these observations suggest that we may approximate temporal shifts as label shifts for comparisons to our bounds.
>
> In addition, our results also now include an evaluation of our bound on violation of equal opportunity subject to covariate shift, in response to Reviewer fUG9’s request for such an example, in the appendices. We note that there are no such bounds for equal opportunity subject to label shift, which preserves equal opportunity (see first paragraph of Section 5).
>
> Our graphics show that our bounds are not vacuous even when the assumptions they depend on are only approximately satisfied. For geographic shifts, the covariate shift  EO bounds (added in revision to Appendix E.2) correctly overestimate disparity and tighten near accurate policies, while our DP bounds are useful only for a subset of policy thresholds. For temporal shift, the label shift bound for DP correctly overestimates the real change of DP violations but still remains at the same order of magnitude. These discussions have been highlighted in line 334–342 at the end of section 7.
>
> Despite our use of the special-case bounds for empirical comparisons, we reassert that our framework is not limited to such assumptions (nor any particular model of distribution shift). While we have focused on these special cases as familiar, worked examples, the development of more general distribution shift bounds within our framework remains promising for future work, as we mention in revision at lines 349-354.
>
> ## 4. Notation (All Reviewers):
>
> All of our reviewers remarked on our dense use of notation.
>
> In part, our notation is complicated by the simultaneous desire to call attention to the unifying statistical structure of distribution shifts and statistical group-disparities and to make our variables consistent with existing literature related to domain adaptation and distribution shifts in machine learning. The appendices (now included in the supplementary material) feature a table of notation (see Table 1: Primary Notation on Section A) that we believe will benefit future readers.

---

> > ### Author Response · Authors · 2022-08-02
> > **General Rebuttal, Part 3/3**
> >
> > ## 5. Minor Errors and Appendix:
> >
> > This omission of the Appendices was a mistake, unnoticed on initial upload; We have included the appendices in revision and apologize for any confusion caused. We have also addressed the typos, minor errors pointed out by the reviewers in the revised version of our paper.
> >
> > ## 6. Summary of Revisions
> >
> > + Include appendix (Reviewer fUG9 and ifHr).
> > + Cite and discuss related work on dynamical fairness and robust training to guarantee fairness (Reviewer k7a3 and fUG9). Lines 61 – 68, 82 – 88; 530 – 534.
> > + Use the conditional independence test tool to decide whether our covariate shift or label shift bounds are appropriate for empirical data (Reviewer k7a3). Lines 321 – 325.
> > + Discuss how to bound the fairness change when both covariate and label distributions shift. (Reviewer ifHr). Lines 349 – 354.
> > + Add intuition for Theorem 3.2 (Reviewer ifHr). Line 191 - 194
> > + Address typos, minor errors (e.g. define premetric, explain replicator dynamics), and add descriptions of figures. (All reviewers)
> > + Add descriptions of our comparisons to empirical data in the abstract (Reviewer fUG9). Line 15 – 16
> > + Provide additional examples and an evaluation of bounds for EO subject to covariate shift (Reviewer fUG9). Line 583 – 590
> > + Provide additional explanations and justifications for our empirical results (All Reviewers). Lines 334 – 342
> > + Discuss how our results may be used in practice in Section 1.2 (Reviewer k7a3, 84wj, ifHr). Line 90 – 100
> >
> >
> > [1] Ensuring Fairness Beyond the Training Data
> > https://proceedings.neurips.cc/paper/2020/hash/d6539d3b57159babf6a72e106beb45bd-Abstract.html
> >
> > [2] Ensuring Fairness under Prior Probability Shifts
> > https://arxiv.org/abs/2005.03474
> >
> > [3] Sample Selection for Fair and Robust Training
> > https://proceedings.neurips.cc/paper/2021/hash/07563a3fe3bbe7e3ba84431ad9d055af-Abstract.html
> >
> > [4]Transferring Fairness under Distribution Shifts via Fair Consistency Regularization
> > https://arxiv.org/abs/2206.12796
> >
> > [5] Delayed Impact of Fair Machine Learning
> > https://arxiv.org/abs/1803.04383
> >
> > [6] Causal Modeling for Fairness in Dynamical Systems
> > https://arxiv.org/abs/1909.09141
> >
> > [7] Maintaining fairness across distribution shift: do we have viable solutions for
> > real-world applications?
> > https://arxiv.org/pdf/2202.01034.pdf

---

### Meta-Review · Area_Chair_ymht · 2022-08-30

**Recommendation:** Accept
**Confidence:** Certain

**Metareview:**

Reviewers agreed that this work, looking at the degradation of group fairness metrics under a few types of distribution shift, add to the literature in this space.  The author's/authors' rebuttal was especially effective in belaying concerns from two reviewers, in addition to many of my own.  I agree with Reviewer 84wj's skepticism of the practical implementability of some theoretical results (e.g. verification of the Lipschitz condition for Thm 3.2), but these are not showstopper concerns; still, please do address them in a final/next version.  Overall, though, the paper addresses a timely topic in a strong and reasonably complete way.

**Award:**

No

---

### Decision · Program_Chairs · 2022-09-14

Accept